# The Effects of Complementary Therapies on Patient-Reported Outcomes: An Overview of Recent Systematic Reviews in Oncology

**DOI:** 10.3390/cancers15184513

**Published:** 2023-09-11

**Authors:** Marit Mentink, Daniëlle Verbeek, Janneke Noordman, Anja Timmer-Bonte, Ines von Rosenstiel, Sandra van Dulmen

**Affiliations:** 1Nivel (Netherlands Institute of Health Services Research), Otterstraat 118, 3512 CR Utrecht, The Netherlands; j.noordman@nivel.nl (J.N.); s.vandulmen@nivel.nl (S.v.D.); 2Department of Primary and Community Care, Radboud Institute for Health Sciences, Radboud University Medical Center, Geert Grooteplein Zuid 21, 6525 EZ Nijmegen, The Netherlands; 3Praktijk Integratieve Oncologie, Heesterpoort 18, 9713 KZ Groningen, The Netherlands; integratieve.oncologie@gmail.com; 4Department of Medical Oncology, Radboud University Medical Center, Geert Grooteplein Zuid 10, 6525 GA Nijmegen, The Netherlands; anja.timmer-bonte@radboudumc.nl; 5Department of Integrative Oncology, Rijnstate Hospital, Wagnerlaan 55, 6815 AD Arnhem, The Netherlands; ivonrosenstiel@rijnstate.nl; 6Faculty of Caring Science, University of Borås, Work Life and Social Welfare, Allégatan 1, 501 90 Borås, Sweden

**Keywords:** complementary medicine, oncology, patient-reported outcomes (PROMS), overview of systematic reviews, meta-analysis

## Abstract

**Simple Summary:**

Complementary therapies, such as acupuncture, yoga and herbal medicine are used by many patients with cancer to relieve symptoms and side effects from anti-cancer treatment. Clinicians in oncology often lack the knowledge to adequately advise patients about the potential benefits and risks of complementary therapies. This study aims to provide an overview of recently published systematic reviews of the effects of complementary therapies on patient-reported health outcomes in patients with cancer. One hundred systematic reviews were included, and the results suggest that several complementary therapies can improve patient-reported health outcomes in patients with cancer. For instance, acupuncture to relieve pain and yoga to improve cancer-related fatigue. The results remain inconclusive for other complementary therapies. Many included systematic reviews did not assess the impact of bias on their results. Nonetheless, the results from this study can help clinicians to find information about the evidence base for complementary therapies when counselling patients on this topic.

**Abstract:**

Many patients with cancer make use of complementary medicine alongside conventional medicine, but clinicians in oncology often lack the knowledge to adequately advise patients on the evidence base for complementary therapies. This study aims to provide an overview of recently published systematic reviews that assess the effects of complementary therapies on patient-reported health outcomes in patients with cancer. Systematic reviews, including a meta-analysis of at least two randomized controlled trials, were identified from the PubMed, Embase, Cochrane Library, CINAHL and PsycINFO databases. The methodological quality was assessed with AMSTAR 2. One hundred systematic reviews were included. The results suggest that several complementary therapies can improve health outcomes reported by patients with cancer, such as acupuncture to relieve pain, music interventions to reduce anxiety and yoga to improve cancer-related fatigue. The side effects related to complementary therapy use are generally mild. The results remain inconclusive for some intervention–outcome combinations. Many of the included systematic reviews insufficiently assessed the causes and impact of bias in their interpretation of the results. This overview of systematic reviews can support clinicians in counselling their patients on this topic and provide directions for future research and clinical practice guidelines in the field of complementary medicine.

## 1. Introduction

Complementary medicine refers to a variety of healthcare interventions not typically part of conventional medical care, but used alongside it. It constitutes of a broad range of modalities, such as mind–body therapies, acupuncture and massage therapy. Complementary medicine use increases when an individual is diagnosed with cancer [1,2]. On average 51 percent of all patients with cancer use complementary medicine [3]. Research shows a variety of motivations for patients with cancer to use complementary medicine, such as treating the side effects of therapy or improving general health [3,4]. In addition, COVID-19-related anxiety and fear seems to have resulted in more complementary medicine use among patients with cancer [5].

Despite the frequent use of complementary medicine by patients with cancer, clinical resources to inform healthcare providers about complementary medicine are scarce. Healthcare providers report limited knowledge about the evidence and potential risks to support complementary medicine use in cancer care [4]. In addition, patients express the need for reliable information about what complementary therapies are helpful when diagnosed with cancer [4]. They often perceive healthcare providers as uninformed or uninterested in complementary medicine [4,6,7], contributing to prevalent non-disclosure of complementary medicine use in patient–provider interactions [8]. Both patient and clinician visit satisfaction was found to increase in consultations in which complementary medicine was discussed [9].

Considering the multitude of complementary therapies and the rapid increase in biomedical publications, it is almost impossible for healthcare providers to keep up with up-to-date evidence on the topic. Existing overviews of systematic reviews for complementary therapies in patients with cancer often focus on one specific intervention [10], specific outcome measures [11] or specific cancer patient populations [12]. The current study aims to provide a comprehensive overview of up-to-date evidence on the effects of complementary therapies on physical, psychological and general patient-reported health outcomes in patients with cancer. Such an overview can facilitate clinical decision-making and support patient–provider communication about complementary medicine.

## 2. Materials and Methods

A review of systematic reviews was performed by using the Cochrane Handbook for Systematic Reviews of Interventions [13]. The protocol is registered on PROSPERO (CRD42022321732). This review is part of a larger mixed methods research project titled ‘COMMON’ [14].

### 2.1. Search Strategy

We systematically searched the PubMed, Embase, Cochrane Library, CINAHL and PsycINFO databases for systematic reviews with a meta-analysis of randomized controlled trials (RCTs) that assess the effect of a complementary therapy on patient-reported outcome measures (PROMs) in patients with cancer. The following main search terms were used: “neoplasm”, “cancer”, “tumor” AND “integrative medicine”, “integrative oncology”, “complementary therapies”, “complementary medicine”, “complementary and alternative medicine” AND “systematic review”, “meta-analysis” (see Appendix A for a full list of the search terms). The PubMed search term was adapted for use with other bibliographic databases.

We included studies published from 1 January 2018 through to 20 April 2022 in the English language. Initially, we searched for records published between 2000 and 20 April 2022, but we decided to narrow the search window due to the large amount of publications for full text screening that were left after the title/abstract screening (*n* = 735). The cutoff point for the year 2018 is based on a comparable overview of systematic reviews published till January 2018 [15]. However, the current review of reviews should not be interpreted as a formal update, because the aforementioned review included only four outcome measures (quality of life, overall survival, pain and depression) and the current review included all patient-reported health outcomes.

### 2.2. Study Selection

The study selection for this review of reviews was conducted according to the PICOS (population, intervention, control, outcomes, study design) framework:

Population: Adults (>18 years) with cancer or cancer survivors. Excluded were systematic reviews that included a population broader than only patients with cancer (except when patients with cancer were separately analyzed).

Intervention: Individual complementary therapies provided alongside conventional cancer treatment or after conventional cancer treatment listed under the MeSH term ‘Complementary Therapies’ in PubMed. Excluded were systematic reviews reporting on: (1) intervention groups (e.g., mind–body therapies) or combined interventions (e.g., acupuncture + yoga) that were not analyzed separately; (2) interventions that were used as alternative, curative therapy (instead of conventional cancer treatment); (3) interventions that are often part of conventional oncology care in Western countries (e.g., psychosocial support, physical therapy, nutritional advice); (4) non-provided therapies (e.g., self-care, exercise, nutrition); (5) separate ingestible or injectable compounds (e.g., herbs, supplements).

Control: Any control group (e.g., usual care, placebo, no treatment, active control).

Outcomes: The effect on patient-reported outcomes (PROMs) considering subjective physical health (e.g., pain, fatigue), psychological health (e.g., depression, anxiety) or general health (e.g., quality of life). Excluded were systematic reviews exclusively reporting on objective health outcome measures (e.g., heart rate, survival rate) or grouped outcomes (e.g., psychological symptoms).

Study design: Systematic reviews published between January 2018 and 20 April 2022 in the English language, including a meta-analysis of at least two RCTs per eligible intervention–outcome combination and reporting effect sizes. Excluded were systematic reviews: (1) including only descriptive, qualitative or case studies as primary studies; (2) providing only a qualitative synthesis of RCTs.

### 2.3. Data Selection

After removal of the duplicates from the combined search results across the databases, all the titles and abstracts were screened in Covidence against the in- and exclusion criteria by two independent reviewers (MM and DV). The full texts of possible eligible studies were retrieved and screened by the same two reviewers (MM and DV). At both stages, disagreements over eligibility were resolved by consensus or by the involvement of the research team (SvD, JN, ATB). During full text screening, it was noticed that the definitions of complementary therapies appeared heterogeneous. Therefore, the reviewers paid additional attention to assessing whether the intervention description of the primary studies included in the systematic review fit our eligibility criteria.

### 2.4. Data Extraction

The data relevant for the purpose of this review were extracted from the systematic reviews by one reviewer (MM) in a data extraction table in Microsoft Word (see Appendix A). The following data were collected and summarized:Descriptive characteristics of the systematic reviews and the included primary studies (author, year, aim, participants, intervention, comparator, outcomes, outcome measures, number of primary studies (RCTs), number of participants in the RCTs).Quantitative outcome data (meta-analysis effect sizes of the eligible outcome–intervention combinations) and the overall conclusion.

In some cases, only part of the systematic review complied with our in- and exclusion criteria on population, intervention, outcome or study design (e.g., a systematic review includes both patient-reported and objective health outcomes). When the complying part of the systematic review (e.g., the patient-reported outcome measures) was separately analyzed, only this part of the data was extracted from the systematic review. If more than one outcome measure was used for a patient-reported outcome (e.g., fatigue status and fatigue score), all outcome measures were extracted but only the outcome measure with the most underlying RCTs was reported in the results section.

### 2.5. Data Synthesis

Given the high heterogeneity in the population, intervention, comparators and outcomes, among the included systematic reviews, pooled meta-analyses were deemed impossible. Therefore, we provide a descriptive synthesis of the quantitative outcome data (meta-analysis) per complementary therapy.

### 2.6. Quality Appraisal

Two reviewers (JN, MvG) appraised the methodological quality of the included systematic reviews according to the seven critical domains of AMSTAR 2 (a measurement tool to assess systematic reviews) [16] (see Appendix A). Disagreements were resolved by the involvement of a third reviewer (MM).

## 3. Results

### 3.1. Study Characteristics

After full text screening for eligibility, 100 systematic reviews were included in the overview (Figure 1). A detailed overview of the included studies and extracted data can be found in Appendix A.

Of the total 100 included systematic reviews, 51 focused on a patient population with various primary cancer diagnoses, 33 focused on patients with breast cancer and 7 on patients with lung cancer. The remaining systematic reviews focused on patients with colorectal, rectal, liver, ovarian or pancreatic cancer, or leukemia. The three most frequently analyzed complementary therapies were herbal medicine, acupuncture and yoga. A description of the complementary therapies can be found in Table 1. Quality of life, fatigue and pain were the most common patient-reported outcome measures.

### 3.2. Methodological Quality

As shown in Figure 2, only half of the systematic reviews included an explicit statement that the methods were established prior to conducting the review and the review protocol was registered or published (item 2). Most included systematic reviews used a satisfactory search strategy, but scored only a Partial Yes because they did not incorporate at least one of the following sources in their search strategy: reference lists, study registries, grey literature or did not consult experts in the field (item 4). Only four systematic reviews provided a list of excluded studies with justifications for their exclusion (item 7). In all but two cases, the risk of bias for individual studies included in the review was assessed (item 9). Although most authors justified combining data and used appropriate weighted techniques to combine study results, the causes for heterogeneity (if present) were not (properly) investigated in several systematic reviews (item 11). In the majority of the systematic reviews, the authors did not account for the risk of bias in individual studies, when interpreting the results in the discussion section (item 13). In almost all the systematic reviews, statistical tests were used for publication bias. However, a discussion on the impact of publication bias on the results was regularly missing (item 15). None of the included systematic reviews scored ‘Yes’ on the seven domains of AMSTAR 2. Only five systematic reviews (partially) met six out of the seven criteria and fourteen systematic reviews (partially) met five out of the seven criteria. The scores on the seven critical domains of AMSTAR 2 for each included systematic review can be found in Appendix A.

### 3.3. Efficacy of Complementary Therapies

The meta-analysis results for the 12 included complementary therapies are shown in Table 2 through to 10 and are described in the text. To maintain the readability of the tables, the patient-reported outcomes with only a few included meta-analyses are not displayed in the tables but in **bold text**.

#### 3.3.1. Acupuncture

Acupuncture seems to have the capacity to alleviate several physical symptoms in patients with cancer (see Table 2). Significant reductions in cancer pain, including aromatase inhibitor-induced arthralgia (AIA) [22,23] and lymphedema pain [24], post-mastectomy pain [25] and neuropathy pain [19,26] were reported in the included meta-analyses.

Peripheral neuropathy symptoms measured with the FACT-NTX (Functional Assessment of Cancer Therapy-Neurotoxicity) subscale seem to be significantly relieved in cancer patients receiving acupuncture [19,26]. However, this effect was not significant in a meta-analysis including only breast cancer patients [27]. When neuropathy symptoms were measured with a variety of instruments in patients with breast cancer, a significant decrease in the symptoms was reported [23].

The side effects of three-step analgesics, such as **nausea**, **vomiting**, **constipation** and **dizziness**, were significantly reduced in patients using acupuncture compared to using only three-step analgesic drugs [17]. In patients with breast cancer, the beneficial effects of acupuncture on gastrointestinal symptoms were reported when RCTs mainly conducted in China were meta-analyzed [23]. No significant effect was reported in a meta-analysis with RCTs originating mainly from Western countries [28].

The meta-analysis results on the effect of acupuncture on hot flash frequency in patients with breast cancer were inconsistent. In one case, a significant effect was reported during acupuncture treatment but not at post-treatment or follow-up [29]. Some meta-analyses indicated a significant reduction in hot flash severity [30] or hot flash scores [27], but not hot flash frequency. A meta-analysis [31], which primary studies entirely overlap with two smaller meta-analyses [28,32], shows a significant negative effect of acupuncture on the incidence of menopausal symptoms measured with the Kupperman index.

Regarding sleep disturbance and quality of life, different measurement instruments were used across the meta-analyses and the results are inconclusive.

Considering psychological patient-reported outcomes, no significant effects were reported in meta-analyses assessing the effect of acupuncture on **anxiety**, **depression** and **cognitive function** [27,30,33]. One meta-analysis of two RCTs reports a significant negative effect of acupuncture on anxiety in patients with breast cancer, but in a pooled analysis of only high-quality articles the effect was not significant [27].

**Xerostomia** (dry mouth) symptoms were significantly relieved in patients receiving acupuncture, according to a meta-analysis of two RCTs [34].

The **functional status**, measured with the Karnofsky Performance Scale (KPS), was significantly improved in the acupuncture combined with opioids group, compared to opioids alone, in a meta-analysis including four RCTs [35].

Furthermore, acupuncture may be effective for improving the quality of life in patients with colorectal cancer by improving **abdominal pain**, **stool score**, **defecation dysfunction**, **sexual dysfunction** and **self-feelings** [36].

**Table 2 cancers-15-04513-t002:** Summary of the meta-analysis results for the effect of acupuncture per outcome (numbers indicate the amount of RCTs in the meta-analysis).

Author and Year	Population	Pain	Neuropathy	Gastrointestinal Symptoms	Fatigue	Hot Flash Frequency	Menopausal Symptoms	Sleep Disturbance	Quality of Life
Chan 2021 [23]	Breast	4 * ^1^	2 *	6 *					9 *
Chien 2020 [31]	Breast					4	5 *		
Gao 2021 [24]	Breast	2 *							
Jang 2020—2 [32]	Breast					3	3		
Kannan 2022 [25]	Breast	2 *							
Li 2021—1 [30]	Breast	3 * ^2^			4 * ^2^	3		2	
Liu 2020 [37]	Breast					3			
Liu 2021 [22]	Breast	3 * ^1^							
Wang 2018 [29]	Breast					3 * ^3^			
Yuanqing 2020 [28]	Breast	5 *		5	4 *	7 *	3		8
Zhang 2021 [27]	Breast	4 *	2		6 *	4		4 *	2 *
Zhu 2021 [33]	Breast	4 *						2	
Chien 2019 [26]	Mixed	3 *	4 *						
Dai 2021 [38]	Mixed	7 *							
Dong 2021 [39]	Mixed	12 *							
He 2020 [40]	Mixed	7 *							
Hou 2020 [35]	Mixed	18 *							
Jang 2020—1 [41]	Mixed				4 *				
Jin 2020 [19]	Mixed	4 *	3 *						
Li 2021—2 [17]	Mixed	18 *							
Lin 2019 [42]	Mixed								3
Tan 2021 [43]	Mixed				2				
Zhang 2018 [44]	Mixed				5 *				

* Asterisk = significant effect; no asterisk = no significant effect. ^1^ Outcome ‘aromatase inhibitor-induced arthralgia’ (AIA). ^2^ Significant effect when compared with waitlist control or usual treatment, non-significant when compared with sham acupuncture. ^3^ Significant effect during treatment, but not post-treatment or at follow-up.

#### 3.3.2. Acupressure

As shown in Table 3, acupressure seems to be effective for the amelioration of physical symptoms in patients with cancer, such as pain (remission rate) [45] and gastrointestinal symptoms, such as nausea, vomiting, constipation and diarrhea [46,47,48]. No significant effects of acupressure were reported on **headache** and **abdominal distension** [46].

#### 3.3.3. Moxibustion

According to the four included meta-analyses, moxibustion can significantly reduce physical symptoms in patients with cancer, such as fatigue [51] (see Table 4). Except for **inappetence** and **abdominal pain**, significant effects on the amelioration of gastrointestinal symptoms, such as nausea and vomiting, constipation, diarrhea and **abdominal distension**, were reported for patients using moxibustion [52,53]. Compared to pneumatic circulation, moxibustion was found to be significantly effective for reducing lymphedema swelling in breast cancer patients [24].

#### 3.3.4. Herbal Medicine

The effect of herbal medicine on gastrointestinal symptoms is often meta-analyzed (see Table 5). The included meta-analyses showed significant negative effects of herbal medicine on the incidence of nausea and vomiting and diarrhea [54,55,56,57,58,59,60,61,62]. **Inappetence** was significantly relieved in patients with primary liver cancer using Chinese medicinal formulas [57]. In patients with colorectal cancer, **anorexia**, **constipation** and **abdominal pain** were not found to be significantly alleviated by the use of traditional herbal medicines, but **abdominal distension** was [55]. No significant effect of Chinese herbal medicine (CHM) was reported on **constipation** experienced by patients with breast cancer [54].

In addition, herbal medicine is frequently used to treat skin-related problems in patients with cancer. For instance, Chinese herbal medicine seemed to exhibit clinical effectiveness on the treatment of epidermal growth factor receptor (EGFR) inhibitor-induced **skin rash** in patients with cancer [63], among which were patients with lung cancer [60]. A meta-analysis including two RCTs showed a significant reduction of chemotherapy-associated **alopecia** in breast cancer patients using Chinese herbal medicine [54]. In patients with pancreatic cancer receiving gemcitabine or docetaxel-based chemotherapy, traditional medicine preparations did not significantly reduce the incidence of **hair loss** [64]. A meta-analysis including 21 RCTs showed that the addition of herbal medicine to fluoropyrimidines therapy was associated with a statistically significant decrease in all-grade **hand–foot syndrome**, measured with three different grading scales [65]. However, this negative effect of herbal medicine was not significant in a meta-analysis in which all-grade hand–foot syndrome was measured with the National Cancer Institute (NCI) criteria [66].

**Oral mucositis** seemed to be significantly alleviated by the use of traditional herbal medicine in patients with colorectal cancer treated with fluoropyrimidine-based chemotherapy [55]. In a meta-analysis including 7 RCTs, CHM appeared to alleviate **depressive symptoms** in cancer patients, either compared with no treatment, antidepressants or psychological treatment [67]. **Menopausal symptoms**, such as hot flashes, depression and irritability, were found to be significantly relieved by the use of CHM in patients with breast cancer treated with endocrine therapy [68].

**Table 5 cancers-15-04513-t005:** Summary of the meta-analysis results for the effect of herbal medicine per outcome (numbers indicate the amount of RCTs in the meta-analysis).

Author and Year	Population	Gastrointestinal Symptoms	Diarrhea	Nausea and Vomiting	Sleep Quality	Fatigue	Pain	Neuropathy	Functional Status	Quality of Life
Bai 2022 [69]	Breast	5 *								12 *
Li 2020 [54]	Breast		4 *	28 * ^1^						
Li 2021 [68]	Breast						3 *		4 *	4 *
Shi 2021 [70]	Breast	9 *							6 *	
Chen 2021 [55]	Colorectal	20 * ^2^	13 *	17 *						
Liu 2019 [66]	Colorectal							9 *		
Li 2020 [56]	Gastric			8 *					4 *	
She 2021 [57]	Liver			7 *					11 *	
Chen 2020 [71]	Lung	14 *							9 *	
Jin 2021 [58]	Lung			5 *						
Kwon 2021 [72]	Lung								5 *	2 *
Lu 2021 [62]	Lung		8 *						2 *	
Wang 2020 [73]	Lung								4 *	
Yang 2020 [59]	Lung		3 *	9 *						5 *
Zhang 2018 [60]	Lung		15 *	3 *		3 *			28 *	
Chen 2019 [63]	Mixed									
Deng 2018 [65]	Mixed								8 *	4 *
Huang 2020 [74]	Mixed					8 * ^3^			4 *	
Li 2019 [75]	Mixed							15 *		
Lin 2019 [42]	Mixed									4 *
Wang 2021 [76]	Mixed						14 *			7 *
Wu 2019 [77]	Mixed	2							4 *	
Yoon 2021 [78]	Mixed				6 *					
Zhao 2020 [79]	Mixed					10 *				
Wang 2019 [61]	Ovarian		2	7 *				3	9 *	3 *
Hu 2022 [64]	Pancreatic	5 *		5					4 *	9 *

* Asterisk = significant effect; no asterisk = no significant effect. ^1^ Nausea and vomiting (toxicity grade III–IV) was significantly reduced, but nausea and vomiting (toxicity grade 0–II) was significantly aggravated in the experimental group receiving herbal medicine. ^2^ Only significant in the subgroup of RCTs without double-blind design. ^3^ Outcome ‘fatigue status’. A non-significant, negative effect was reported for the outcome ‘fatigue score’.

#### 3.3.5. Mindfulness-Based Stress Reduction (MBSR)

MBSR is mainly used for the treatment of psychological symptoms in patients with cancer (see Table 6). Significant effects of MBSR on the amelioration of depression were reported [80,81,82]. According to three meta-analyses, anxiety was significantly relieved by MBSR [80,81,82,83]. In patients with breast cancer, the meta-analysis results on the effect of MBSR on anxiety and sleep quality were inconclusive [80,81]. A significant effect on pain relief and improvement of the quality of life were reported in a meta-analysis of MBSR in a mixed cancer patient population [82]. In patients with breast cancer receiving MBSR, no significant effects were reported on the outcomes for pain and quality of life [80].

#### 3.3.6. Music Interventions

Music interventions can be used to relieve a variety of symptoms in patients with cancer, especially psychological symptoms such as depression, anxiety, low mood and distress (see Table 7). For instance, two meta-analyses reported significant effects of music interventions for the relief of depression [87,88]. In addition, significant effects were reported for music interventions on the improvement of cancer pain [88,89,90], sleep quality [87], fatigue [91] and quality of life [87,88].

#### 3.3.7. Tai Chi

The results of the included meta-analyses indicated that Tai Chi can decrease cancer-related fatigue [92,93,94,95] (see Table 8). Considering the psychological symptoms, a meta-analysis of two RCTs showed that Tai Chi can alleviate anxiety [93,96]. There was no evidence that Tai Chi decreases the symptoms of depression [96,97]. Sleep quality was found to be improved by Tai Chi in one meta-analysis including RCTs with different cancer types [94], but a non-significant positive effect was reported for patients with breast cancer [97]. A significant positive effect was found for Tai Chi on quality of life [93,94,97].

#### 3.3.8. Qigong

Qigong seems a promising intervention for the relief of cancer-related fatigue [98,99,100] (see Table 9). Sleep quality was significantly improved by Qigong according to a meta-analysis of two RCTs [98]. One meta-analysis showed a significant positive effect of Qigong exercise on quality of life [98], whilst a meta-analysis including different RCTs showed a non-significant positive trend [42].

#### 3.3.9. Yoga

The effect of yoga on cancer-related fatigue is often meta-analyzed and the results show a significant decrease in fatigue [92,101,102,103,104,105,106,107] (see Table 10). In addition, yoga seems effective for the management of psychological symptoms in patients with cancer, such as anxiety, depressive symptoms and stress [101,104,107,108,109]. A significant reduction in sleep disturbance in the yoga intervention group was reported [104,107,110]. However, a meta-analysis including seven RCTs showed that yoga did not significantly improve sleep quality measured with the Pittsburgh Sleep Quality Index (PSQI) in patients with breast cancer [111]. In a meta-analysis including two RCTs the effect of Yoga on **cognitive impairment** was studied, but did not report a significant effect [112].

#### 3.3.10. Art Therapy

In one meta-analysis including two RCTs on expressive writing, a significant negative effect of expressive writing on depressive symptoms was reported in patients with breast cancer [108] (see Appendix A).

#### 3.3.11. Manual Therapy

Two meta-analyses assessed the effect of manual lymphatic drainage (MLD) in breast cancer patients with lymphedema and yielded contradictory results: the first meta-analysis included three RCTs and showed a significant negative effect on lymphedema pain [115]. The second meta-analysis included two RCTs, of which one overlapped with the first meta-analysis, and showed a non-significant negative effect of MLD on pain [116].

A meta-analysis among patients with breast cancer revealed a significant negative effect of manual therapy (e.g., massage, myofascial release) on chronic musculoskeletal pain [117]. A significant effect of myofascial release on the relief of post-mastectomy pain was reported in a meta-analysis including two RCTs [25].

Manual therapies did not seem to have a significant positive effect on quality of life [42,115,117].

#### 3.3.12. Relaxation Therapy

One meta-analysis [118] evaluated the effect of the combined practice of progressive muscle release and guided imagery in patients with breast cancer. Significant effects were reported on the improvement of the quality of life (two RCTs) and the amelioration of stress (five RCTs), anxiety (five RCTs), depression (five RCTs) and nausea and vomiting (two RCTs).

One meta-analysis [119] reported that hypnosis before general anesthesia for breast cancer surgery could significantly reduce anxiety (six RCTs) and post-operative pain (seven RCTs), but not post-operative nausea and vomiting in patients with breast cancer undergoing minor surgery.

### 3.4. Safety of Complementary Therapies

The majority of the included systematic reviews (60 out of 100) reported on the incidence or absence of adverse events related to complementary therapy use. Reported adverse events were generally mild, such as bruising related to acupuncture treatment [23], cramps related to yoga practice [102] or a skin allergy related to herbal medicine use [75]. In the included systematic reviews on herbal medicine, potential herb–drug interactions were not systematically reported. Only a few included systematic reviews included an explicit statement about the interaction effects [54,63,64,72], such as insufficient documentation and monitoring of traditional medicine preparations by clinicians and the lack of (pharmacokinetic) studies on herb–drug interactions.

In many cases, the group of patients receiving anti-cancer treatment combined with a complementary therapy experienced less adverse reactions from anti-cancer treatment compared to the control group receiving only anti-cancer treatment (for example: [35,64,67,74]). In two instances, a significant deteriorating effect of a complementary therapy on a patient-reported outcome was reported. First, in the group of breast cancer patients receiving herbal medicine and chemotherapy, the frequency of non-severe (grade 0-II) nausea and vomiting was significantly increased compared to patients receiving only chemotherapy [54]. Nonetheless, severe nausea and vomiting (grade III-IV) frequency was significantly alleviated in the experimental group. Second, sleep quality measured with the Pittsburgh Sleep Quality Index (PSQI) was significantly less improved in the intervention group receiving MBSR compared to the active control (i.e., psychoeducation or stress management techniques) [85]. When compared to usual care, MBSR significantly improved sleep quality.

## 4. Discussion

This study aimed to summarize the evidence on complementary therapies on patient-reported health outcomes in patients with cancer. This resulted in an overview of one hundred recently published meta-analyses, which described the effects of twelve different complementary therapies on several health outcomes reported by patients with cancer. To the best of our knowledge, such a comprehensive overview of the systematic reviews on this topic is not available yet.

According to the results from this overview of systematic reviews, some complementary therapies can ameliorate physical and psychological symptoms caused by cancer itself or its treatment and, importantly, improve the quality of life as reported by patients with cancer. Side effects from complementary therapies barely occurred, and if they occurred, they were generally mild. Despite diversity in the methodological quality of the included meta-analyses, the selected complementary therapies show beneficial effects on patient-reported health outcomes. For instance, the included meta-analyses show the efficacy of acupuncture and related therapies to relieve cancer pain in different subgroups of patients and yoga to improve cancer-related fatigue. The use of acupuncture for pain management was also recommended in a recently published clinical practice guideline on integrative medicine in oncology [120]. The results from the current overview seem consistent with several other existing clinical recommendations, such as music interventions to reduce anxiety [121], acupuncture to relieve neuropathy symptoms [120,122] or yoga to improve quality of life [123].

For some intervention–outcome combinations, the results were inconclusive. This is probably due to variations in the systematic review’s eligibility criteria on the study population, intervention (delivery), comparators or outcome measures that are inherent in such a comprehensive overview of systematic reviews. It was beyond the scope of this study to analyze the differences within an intervention. For instance, the enormous amount of different botanical compounds used in herbal medicine can easily lead to inconclusive results on the same patient-reported outcome. Variations in the measurement instruments used for the same patient-reported outcome between meta-analyses may also lead to inconclusive results.

Only a few systematic reviews met most of the criteria, as defined in the seven critical domains of AMSTAR 2 [16]. However, not all criteria equally compromise the research process quality. For instance, a lack of protocol registration does not mean that the protocol is significantly violated without justification, or the absence of a list of missing studies does not mean that there is no proper justification for exclusion. An important criterion often violated by the systematic review’s authors, is accounting for the impact of the risk of bias (including publication bias) in the interpretation and discussion of the results. In addition, the causes of heterogeneity among combined primary study results were not (properly) investigated in several included meta-analyses. When authors do not take into account these factors when interpreting the meta-analysis results, there is an increased risk of drawing distorted conclusions about the effect of complementary therapies. In almost all the systematic reviews, the authors raised their concerns about the quality of the underlying primary studies and the need for randomized clinical trials with larger sample sizes. When interpreting the results from individual meta-analyses, it is important to consider that the nature of complementary therapies makes it difficult to conduct double-blinded trials. The lack of double blinding makes it more likely to show an advantage of complementary therapy due to a placebo effect or response bias, especially in patient-reported outcome measures.

### 4.1. Limitations

When interpreting the results section, it is important to take the following limitations into account. First, the RCTs of the included meta-analyses often (partially or entirely) overlap because we did not control for the duplication of primary studies. Thus, the amount of included meta-analyses has no value. Second, the meta-analyses effect sizes were not comparable because of heterogeneity in the effect size type and the interpretation of the effect sizes was dependent on the study parameters and clinical outcomes. However, the effect sizes for each included meta-analysis can be found in Appendix A. Third, because of the broad scope of this overview of systematic reviews, we did not differentiate between the patient populations (e.g., active treatment or survivor), intervention delivery (e.g., intensity and duration), comparators and outcomes (e.g., different instruments and measurement moments). However, if the meta-analysis outcomes were significantly affected by one of these factors the information was generally extracted and reported in the current paper.

Lastly, it is possible that relevant meta-analyses for particular complementary therapies, outcomes or in specific cancer populations are missing due to the narrowing of our search window (January 2018–April 2022). For example, two systematic reviews published before 2018 showed beneficial effects of massage therapy on pain in cancer patients [124,125], whilst in the current review only a few systematic reviews on manual therapies were included (e.g., lymphatic drainage, myofascial release). Nonetheless, the included meta-analyses most likely will cover the majority of relevant primary studies on complementary therapies.

### 4.2. Applications

Given the lack of knowledge on complementary therapies reported by clinicians, this up-to-date overview of the evidence can support them in counselling patients with cancer on the topic by describing the effect of complementary therapies on a range of outcomes relevant for patients with cancer. A complicating factor is that evidence on the efficacy of some complementary therapies is inconclusive. As proposed in the ethical framework by Cohen et al. [126], clinicians can tolerate the use of complementary therapies for which evidence is inconclusive, but safety is supported by evidence as long as caution is provided and effectiveness is closely monitored. This way, patient beliefs and decisions about complementary therapy use can be respected in a safe manner. The recently published clinical practice guideline by Balneaves and colleagues [127] provides further recommendations for oncology healthcare providers for addressing complementary medicine use among patients with cancer. For instance, by supporting patients to make evidence-informed decisions on complementary medicine use by ensuring that the patient understands the potential benefits and risks related to this use.

The evidence presented in this study was not pooled and not graded, which means that it was not possible to make explicit clinical recommendations. However, the results can provide leads for updating clinical recommendations. For instance, the included meta-analyses indicated that yoga can relieve fatigue among patients with breast cancer. In the last update to the clinical practice guideline on integrative therapies for breast cancer patients [121], there was insufficient evidence to form a clinical recommendation on this intervention–outcome combination.

### 4.3. Future Studies

The results from the current study can indicate directions for future research in the field of complementary medicine, since this overview clarifies which complementary therapy, outcome measure and in which population little research has been conducted. For example, the included meta-analyses included only two RCTs indicating a positive effect of Tai Chi on anxiety, which is an intervention–outcome combination that should be further explored. Some complementary therapies, such as yoga and Qigong, did not significantly improve patient-reported outcomes when compared to the active control (e.g., regular exercise) [100,101,106]. Non-inferiority trials are recommended.

Many of the described complementary therapies in this overview have their roots in traditional Chinese medicine. A large proportion of the primary studies have been conducted in Chinese clinical research samples. For future research, it would be interesting to investigate to what extent the results are applicable to other populations, such as patients from Western countries.

The safety of complementary therapy use was not assessed in several included systematic reviews, despite its relevance for clinical decision-making [126]. The lack of adverse event reporting in RCTs and SRs affects the reliability of the safety judgment for complementary therapies. Therefore, it is important that future studies always report on (the absence of) adverse events that occur during complementary therapy use and also take into account possible interaction effects with anti-cancer treatment, especially for herbal medicine.

In the included meta-analyses in the current study, it was reported a few times that the effect of a complementary therapy on a patient-reported outcome was only significant during or immediately post-intervention. This finding indicates that the effect of complementary therapies, such as acupuncture and yoga, diminishes once you stop using it. For future research, longer follow-ups are recommended to provide more insight into effect duration, which is valuable information for patients. In addition, longer follow-ups can provide more information about the long-term safety of complementary therapy use.

## 5. Conclusions

This study provides an overview of the recently published systematic reviews that assess the effects of complementary therapies on physical, psychological and general patient-reported health outcomes in patients with cancer. The results suggest that several complementary therapies can have an effect on the improvement of symptoms or side effects from treatment reported by patients with cancer. For example, the included meta-analyses show the effectiveness of acupuncture to relieve pain, yoga to improve fatigue and music interventions to reduce anxiety. Importantly, complementary therapy use in general seems to improve the quality of life for patients with cancer. The side effects related to complementary therapy use are generally mild. For some intervention–outcome combinations, the results remain inconclusive. Rigorous randomized clinical trials on the effect of complementary therapies are warranted. In many of the included systematic reviews, the causes and impact of bias were insufficiently assessed in the interpretation of the results. Nonetheless, this up-to-date overview of the evidence on complementary therapies could support clinicians in counselling their patients on this topic and could provide directions for future research and clinical practice guidelines.

## Figures and Tables

**Figure 1 cancers-15-04513-f001:**
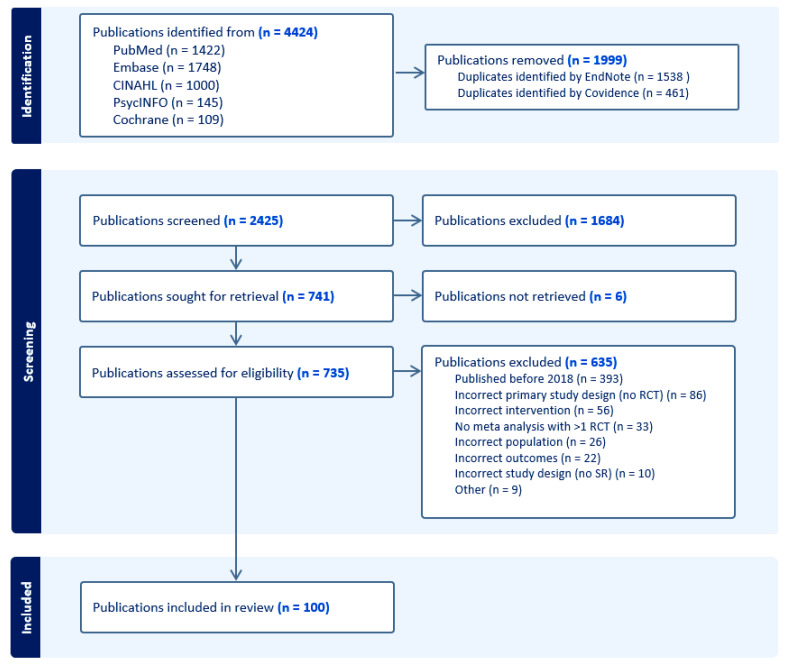
PRISMA flow chart on the identification of the included publications.

**Figure 2 cancers-15-04513-f002:**
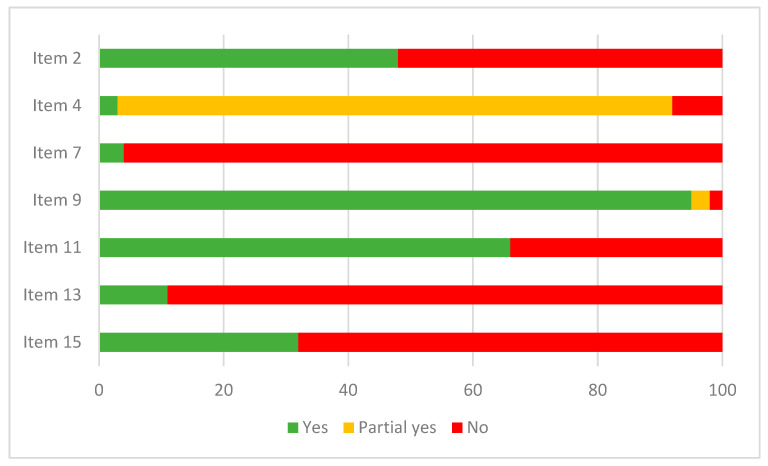
Quality assessment of the systematic reviews according to the seven critical domains of AMSTAR 2 [14].

**Table 1 cancers-15-04513-t001:** Description of the included complementary therapies.

Complementary Therapy (No. Included Meta-Analyses)	Definition
Herbal medicine [17]	A type of medicine that uses roots, stems, leaves, flowers or seeds of plants [18] (we did not include meta-analyses assessing the effectivity of individual herbs)
Acupuncture [19]	The technique of inserting thin needles through the skin at specific points on the body (including electroacupuncture in which pulses of weak electrical current are sent through acupuncture needles into acupuncture points in the skin) [18]
Yoga [16]	An ancient system of practices used to balance the mind and body through exercise, meditation (focusing thoughts), and control of breathing and emotions [18]
Mindfulness-based stress reduction [7]	A moment-to-moment awareness of one’s experience without judgment and as a state and not a trait [20] (we included only mindfulness-based stress reduction (MBSR), a structured group program of mindfulness training [21])
Acupressure [7]	The application of pressure or localized massage to specific sites on the body [18] (including ear acupressure in which seeds or pellets are taped on auricular acupoints)
Tai Chi [6]	One of the martial arts and also a form of meditative exercise using methodically slow circular stretching movements and positions of body balance [20]
Music interventions [5]	A type of therapy that uses music to help improve a person’s overall health and well-being [18] (we included passive and active listening music interventions)
Manual therapy [5]	A type of therapy in which the therapist moves or manipulates one or more parts of the patient’s body [18]
Qigong [4]	An ancient Chinese system of postures, exercises, breathing techniques and meditations designed to improve and enhance the body’s Qi (e.g., vital energy) [20]
Moxibustion [4]	A type of heat therapy in which a herb is burned on or above the skin to warm and stimulate an acupuncture point or affected area [18]
Relaxation therapy [2]	Treatment to improve one’s health condition by using techniques that can reduce physiological stress, psychological stress, or both [20]
Art therapy [1]	Treatment that uses the making of art and the response to art to improve one’s physical, mental and emotional well-being [18]

**Table 3 cancers-15-04513-t003:** Summary of the meta-analysis results for the effect of acupressure per outcome (numbers indicate the amount of RCTs in the meta-analysis).

Author and Year	Population	Pain	Fatigue	Sleep Quality	Nausea and Vomiting	Constipation	Diarrhea	Quality of Life
Chen 2018 [47] ^a^	Leukemia					3 *		2 *
Jing 2018 [48] ^a^	Leukemia					3 *		2 *
Chen 2021 [46] ^a^	Mixed		2 *		10 *	6 *	2 *	
Hsieh 2021 [49]	Mixed		14 *					
Mai 2022 [45]	Mixed	11 *		7 *				8 *
Tan 2021 [43]	Mixed		2					
Wang 2021 [50] ^a^	Mixed			6 *				

* Asterisk = significant effect; no asterisk = no significant effect, ^a^ auricular acupressure.

**Table 4 cancers-15-04513-t004:** Summary of the meta-analysis results for the effect of moxibustion per outcome (numbers indicate the amount of RCTs in the meta-analysis).

Author and Year	Population	Fatigue	Nausea and Vomiting	Constipation	Diarrhea	Lymphedema (Swelling)	Functional Status	Quality of Life
Gao 2021 [24]	Breast					3 *		
Ma 2019 [51]	Mixed	10 *						9 *
Yao 2022 [52]	Mixed		24 *	7 *	7 *		6 *	2 *
Zhang 2018 [53]	Mixed		7 *				4 *	3 *

* Asterisk = significant effect; no asterisk = no significant effect.

**Table 6 cancers-15-04513-t006:** Summary of the meta-analysis results for the effect of MBSR per outcome (numbers indicate the amount of RCTs in the meta-analysis).

Author and Year	Population	Anxiety	Depression	Stress	Fatigue	Sleep Quality	Pain	Quality of Life
Chang 2021 [80]	Breast	3	6 *	3	4 *	5	5	4
Schell 2019 [81]	Breast	6 *	6 *		5 *	4 *		
He 2020 [84]	Mixed				5 *			
Lin 2022 [82]	Mixed	14 *	12 *		8 *		4 *	7 *
Suh 2021 [85]	Mixed					9 * ^1^		
Xie 2020 [86]	Mixed				14 *			
Xunlin 2020 [83]	Mixed	9 *						

* Asterisk = significant effect; no asterisk = no significant effect. ^1^ Significant increase in sleep quality only when compared to the usual care control, mixed results when compared to the active control.

**Table 7 cancers-15-04513-t007:** Summary of the meta-analysis results for the effect of music interventions per outcome (numbers indicate the amount of RCTs in the meta-analysis).

Author and Year	Population	Depression	Anxiety	Mood	Distress	Pain	Sleep Quality	Fatigue	Functional Status	Quality of Life
Bro 2018 [89]	Mixed	2	9 *	4 *	2	9 *		3		2
Li 2020 [88]	Mixed	6 *	6 *			5 *				10 *
Qi 2021 [91]	Mixed							8 *		
Yang 2021 [87]	Mixed	14 *	8 ^1^				4 *		3 *	7 *
Yangöz 2019 [90]	Mixed					6 *				

* Asterisk = significant effect; no asterisk = no significant effect. ^1^ Large, significant effect of music intervention on anxiety measured with the Hamilton Anxiety Rating Scale (HAM-A), but not significant when measured with the Self-Rating Anxiety Scale (SAS).

**Table 8 cancers-15-04513-t008:** Summary of the meta-analysis results for the effect of Tai Chi per outcome (numbers indicate the amount of RCTs in the meta-analysis).

Author and Year	Population	Fatigue	Anxiety	Depression	Sleep Quality	Pain	Quality of Life
Liu 2021 [92]	Breast	2 *					
Liu 2020 [97]	Breast	2 ^1^		3	2		2 *
Luo 2020 [93]	Breast	3 *	2 *			4 *	5 *
Cai 2022 [96]	Mixed		2 *	4			
Ni 2019 [94]	Mixed	3 *			3 *		8 * ^2^
Song 2018 [95]	Mixed	6 * ^3^					

* Asterisk = significant effect; no asterisk = no significant effect. ^1^ Compared to conventional supportive care. Significant decrease in fatigue when Tai Chi was used as an adjunct to conventional care. ^2^ Outcome measure ‘physical functioning’. Psychological functioning also has a medium significant effect. Social functioning not significant. ^3^ Measured post-intervention. At 3-month follow-up, no significant effect of Tai Chi on fatigue.

**Table 9 cancers-15-04513-t009:** Summary of the meta-analysis results for the effect of Qigong per outcome (numbers indicate the amount of RCTs in the meta-analysis).

Author and Year	Population	Fatigue	Sleep Quality	Quality of Life
Kuo 2021 [98]	Mixed	5 *	2 *	3 *
Lin 2019 [42]	Mixed			3
Wang 2021 [99]	Mixed	4 *		
Yin 2020 [100]	Mixed	13 * ^1^		

* Asterisk = significant effect; no asterisk = no significant effect. ^1^ Qigong compared to usual care/waitlist control. No significant effect when compared with Western exercise/treatment control.

**Table 10 cancers-15-04513-t010:** Summary of the meta-analysis results for the effect of yoga per outcome (numbers indicate the amount of RCTs in the meta-analysis).

Author and Year	Population	Fatigue	Anxiety	Depression	Stress	Sleep Disturbance	Pain	Quality of Life
Coutiño 2019 [108]	Breast			5 *				
Dong 2019 [102]	Breast	18 *						
El-Hashimi 2019 [113]	Breast							8
Hsueh 2021 [104]	Breast	14 *	8 *	12 *	4 *	5 *	5 *	5 * ^1^
Liu 2021 [92]	Breast	13 *						
O’Neill 2020 [106]	Breast	18 * ^2^						10 * ^2^
Wang 2020 [111]	Breast					7 ^3^		
Yi 2021 [107]	Breast	4 * ^4^	5 *	6 *		2 * ^4^		3 *
Armer 2020 [101]	Mixed	29 * ^2^		12 * ^2^				17
Danon 2021 [114]	Mixed						7	
Gonzalez 2021 [109]	Mixed		15 *	25 *				
Haussmann 2022 [103]	Mixed	24 *						
Jihong 2021 [105]	Mixed	11 *						
Lin 2019 [42]	Mixed							7
Tang 2019 [110]	Mixed					13 *		

* Asterisk = significant effect; no asterisk = no significant effect. ^1^ Functional, emotional and social well-being were significantly improved by yoga. Effect on physical well-being was not significant. ^2^ Only a significant effect when compared to the active control group. ^3^ Outcome measure ‘sleep quality’. ^4^ Measured post-intervention. No significant effect when measured at medium or long-term follow-up.

## Data Availability

The data can be shared up on request.

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
