# Peer review of "The Effects of Complementary Therapies on Patient-Reported Outcomes: An Overview of Recent Systematic Reviews in Oncology"

_cancers, 2023, doi:10.3390/cancers15184513_

Round 1

Reviewer 1 Report

This manuscript is well written and provides a systematic review of approximately 100 meta-analyses to describe the effects of complementary therapies on several health outcomes reported by cancer patients. However some points should be improved.

- Throughout the paper, intensity of side effects related to complementary therapies is detailed (mild, moderate, medium-to-large sized effects ...). Please also detail the scale used to assess side effects

-  Results, Efficacy of complementary therapies, page 6, line 205                    Although mentioned, the table 3.11 does not appear in the paper. Please add table 3.11.

 - Results, herbal medicines, page 8-9                                                    Despite authors call for caution on drugs interactions with herbal medicines interactions page 15, lines 524-525, this point is almost important and should be detailed in Results. If none of the 100 publications refer to this risk, it should be specified.

 -Results, page 13 line 419                                                                                    Text formatting, incorrect word breaks : man-agement

Discussion, page 13, lines 433-435                                                                    « despite diversity .... seems quite robust » : the study robustness is over-estimated compared to the results. This entire sentence should be deleted or shortened

Reviewer 2 Report

The manuscript provided an overview of recently published systematic reviews of the effects of complementary therapies on patient-reported health outcomes in patients with cancer, which suggest that several complementary therapies can influence the improvement of symptoms or side-effects from treatment reported by patients with cancer. I suggest a major revision.

1. In table 3, over half of the meta-analysis results for effect of Acupuncture involved focus on the patients with breast cancer, indicating a sample data bias of population.

2. The reasons why patient-reported outcomes in bold text are not reported in a Table are not explained.

3. The annotations of some tables are confusing for the meanings of symbols such as asterisk are inconsistent.

4. The cited references need to be significantly improved. Currently, there are several citations that are too old, such as published in 2002. Those references should be deleted or substituted. For example, it is recommended to substitute the #124 reference Cohen, M.H.; Eisenberg, D.M. Potential physician malpractice liability associated with complementary and integrative medical therapies. Ann Intern Med 2002, 136, 596-603.with a recently published manuscript, PMID 27028029.

5. In the Introduction, the authors should report the CAM need of patients. Enen in the covid-19 pandemic, the patients used CAM more. PMID: 37353374could be cited.

6. I would encourage the authors to include these studies below as this would enhance the manuscript.

* Balneaves L G, Watling C Z, Hayward E N, et al. Addressing complementary and alternative medicine use among individuals with cancer: an integrative review and clinical practice guideline. JNCI: Journal of the National Cancer Institute, 2022, 114(1): 25-37.

* Yuan C, Zhang W, Wang J, et al. Chinese Medicine Phenomics (Chinmedphenomics): Personalized, Precise and Promising. Phenomics, 2022, 2, 383388.

* Kim D, Sung S H, Shin S, et al. The effect of cancer on traditional, complementary and alternative medicine utilization in Korea: a fixed effect analysis using Korea Health Panel data. BMC Complementary Medicine and Therapies, 2022, 22(1): 137

7. Usually, some complementary therapies are applied in combination such as the acupuncture and moxibustion while the author ignored related research.

None.

Reviewer 3 Report

The Authors in their work make an update of complementary therapies in patients with cancer.

The topic is emerging and they have tried to summarize the most recent literature ,

I think the paper is okay,

Author Response

Thank you for your feedback. 

Round 2

Reviewer 2 Report

I have no other suggestions.